# Predicting Mandarin Fruit Acceptability: From High-Field to Benchtop NMR Spectroscopy

**DOI:** 10.3390/foods11162384

**Published:** 2022-08-09

**Authors:** Ignacio Migues, Fernando Rivas, Guillermo Moyna, Simon D. Kelly, Horacio Heinzen

**Affiliations:** 1Laboratorio de Farmacognosia y Productos Naturales, Departamento de Química Orgánica, Facultad de Química, Universidad de la República, Montevideo 11800, Uruguay; 2Programa de Posgrados de la Facultad de Química, Universidad de la República, Montevideo 11800, Uruguay; 3Programa Nacional de Investigación en Producción Citrícola, Instituto Nacional de Investigación Agropecuaria (INIA), Salto 50000, Uruguay; 4Laboratorio de Espectroscopía y Fisicoquímica Orgánica, Departamento de Química del Litoral, CENUR Litoral Norte, Universidad de la República, Paysandú 60000, Uruguay; 5Food Safety Control Laboratory, Joint FAO/IAEA Centre of Nuclear Techniques in Food and Agriculture, Department of Nuclear Sciences and Applications, International Atomic Energy Agency, 1400 Vienna, Austria

**Keywords:** acceptability, benchtop NMR, mandarins, NMR

## Abstract

Recent advances in nuclear magnetic resonance (NMR) have led to the development of low-field benchtop NMR systems with improved sensitivity and resolution suitable for use in research and quality-control laboratories. Compared to their high-resolution counterparts, their lower purchase and running costs make them a good alternative for routine use. In this article, we show the adaptation of a method for predicting the consumer acceptability of mandarins, originally reported using a high-field 400 MHz NMR spectrometer, to benchtop 60 MHz NMR systems. Our findings reveal that both instruments yield comparable results regarding sugar and citric acid levels, leading to the development of virtually identical predictive linear models. However, the lower cost of benchtop NMR systems would allow cultivators to implement this chemometric-based method as an additional tool for the selection of new cultivars.

## 1. Introduction

Since the early days of nuclear magnetic resonance (NMR), considerable efforts have been invested to increase sensitivity and spectral resolution through the use of magnets with stronger fields. These endeavors have gone hand in hand with the development of novel superconducting materials and cryogenic technologies [1]. However, these systems are generally expensive and have high running and maintenance costs, driving many NMR spectrometer manufacturers to develop smaller and more accessible systems based on cryogen-free permanent magnets. These low-field instruments have magnetic fields below 2.3 T (i.e., ^1^H resonance frequencies under 100 MHz), fit on a regular laboratory benchtop, and are even suitable for use in field experiments [2,3]. The basis of these instruments is the use of rare-earth ring-shaped magnets that produce relatively strong and homogeneous fields [3]. Their lower sensitivity can sometimes be offset by concentrating the samples or using a variety of methodologies for the enhancement of Boltzmann polarization [4]. Similarly, issues with chemical shift resolution can be addressed through the application of different signal acquisition and processing techniques, including solvent suppression and gradient-based pulse sequences [3,5].

Although low-field benchtop NMR spectrometers may not be suitable for natural product research due to their lower sensitivity and resolution, they have been used successfully in the quality control of phytopharmaceuticals and in food analysis, to mention a few examples [6]. In academia, the use of benchtop NMR is increasing progressively. The low operating costs and ease of use of these instruments allow students not only to control their research products, but to follow chemical reactions in real time or even perform quantitative analyses [6,7,8,9]. Indeed, the quantitation of natural products using benchtop NMR has been employed in the quality control of drugs [10] and for the detection of adulterations in pharmaceutical products [11,12]. The use of low-field NMR in routine quality control of foods has also been demonstrated [6]. Examples of such applications include the determination of alcohol content in beverages [13] and the study of food authenticity and food fraud by targeted and untargeted analysis, where wine, coffee, oils, or even meat are examples [14,15,16,17,18]. For certain products, subdisciplines have been developed to study metabolomic profiles. For example, the term “MEATabolomics” refers to the application of metabolomic analysis to correlate the composition of meat with its sensory attributes [19,20].

In food analysis, untargeted approaches are preferred when trying to discover flavor-related compounds, which are followed with targeted analyses to measure the content of specific compounds or study metabolic pathways of interest [21,22]. Citrus metabolomics has been emerging in the last few years to control industrial processes or to evaluate flavor traits that influence consumer preferences [23,24,25]. However, little research has been conducted to adapt high-field NMR techniques to low-field systems. As stated by Castaing-Cordier and coworkers [26], benchtop instruments can be used in many applications due to recent advances in terms of sensitivity and resolution. Recently, we proved the usefulness of high-field NMR to predict consumer preferences in mandarins. Although interesting from an academic point of view, the high cost of the 400 MHz spectrometer employed in the study hampers its application by the local citrus industry [25]. The aim of the present work is to show an updated protocol for the analysis of mandarin consumer preferences using benchtop NMR systems that could be accessible to citrus fruit cultivators. As shown herein, our results indicate that chemometric-based consumer acceptability models of identical quality can be obtained regardless of magnetic field.

## 2. Materials and Methods

The samples used in the comparisons were a selection of aqueous mandarin extracts obtained during the development of the original method at 400 MHz [25]. Five extract replicates for each mandarin variety were lyophilized and stored under nitrogen in sealed containers until analysis. They were then dissolved in 600 μL of deuterium oxide (MagniSolv™, 99.9% D, Merck, Darmstadt, Germany), transferred to 5 mm NMR tubes (Norell^®^ Standard Series^TM^ Sigma-Aldrich, Darmstadt, Germany) and analyzed immediately.

A Bruker Avance III 400 spectrometer (Bruker, Ettlingen, Germany) was used to perform the high-field NMR experiments, while a Magritek Spinsolve 60 benchtop NMR spectrometer (Magritek GmbH, Aachen, Germany) was used to obtain the data at the low field. The 400 MHz spectra were obtained at a ^1^H frequency of 400.13 MHz using a *z*-gradient BBFO-Plus probe (298 K). Spectra were recorded using a spectral width of 8 KHz, a data size of 32 K, and using a 30° excitation pulse. A total 64 scans with a relaxation delay of 1 s between scans were averaged, leading to an analysis time of 4.1 min per sample. The 60 MHz data were obtained at room temperature using a ^1^H frequency of 62.32 MHz, a spectral width of 5 KHz, a data size of 32 K, and using a 90° excitation pulse. A total of 256 scans with a relaxation delay of 1 s between scans were averaged in this case, resulting in a total analysis time of 64.0 min per sample.

All spectra were processed using MNova (version 11.0, MestreLab Research, S.L., Santiago de Compostela, Spain) following an identical protocol, which included zero filling to 64 K and apodization with a 0.3 Hz exponential window function prior to Fourier transformation, manual phase and baseline correction, and referencing to the signal of the anomeric proton of α-glucose at 5.22 ppm. The spectra were then aligned using the derivative method and the average spectrum as a reference [27].

Once all spectra were aligned, the integral of the signal belonging to the sucrose glucosyl anomeric proton at 5.40 ppm was given an arbitrary value of 1.00. Then, the areas of the signals corresponding to the anomeric protons of α-glucose at 5.22 ppm, β-glucose at 4.63 ppm, and the multiplet arising from the H-3 and H-4 protons of the β-furanose form of fructose at 4.09 ppm, together with the four citric acid methylene protons centered at approximately 2.8 ppm, were scaled to that of the sucrose signal. The integration ranges for the sugar signals mentioned above were, respectively, 5.54 to 5.32, 5.29 to 5.16, 4.63 to 4.53, and 4.10 to 4.07 ppm in both instruments. Due to slight differences in the temperature of the experiments, the citrate signals were integrated from 3.02 to 2.73 ppm in the high-field spectrometer, and between 2.81 and 2.54 ppm on the benchtop instrument.

The relative area values were corrected using the sweetness scale of Schiffman and coworkers [28,29], being 1.0 for sucrose, 1.3 for fructose and 0.6 for α- and β-glucose. The ratio sweetening power/citric acid was calculated as follows, where *n* represents each of the sugars considered:(1)$$\frac{\sum_{n}(Sugar\;sweetness\times Sugar\;content{)}_{n}}{Citric\;acid\;content}$$

The correlation between the mandarin acceptability and the sweetening power/citric acid ratio was determined using the same mandarin varieties for both spectrometer systems, the R^2^ of the regressions was determined and the root mean square error (RMSE) of each model was calculated.

## 3. Results and Discussion

Figure 1 shows spectra obtained at 400 and 60 MHz for the same aqueous extract, respectively. Given its higher resolution, the spectrum obtained at 400 MHz allows for the identification of most protons from the species of interest. On the other hand, several of these signals appear overlapped at 60 MHz, making the initial assignment of resonances a harder task that requires technical know-how.

However, the signals corresponding to the sugar anomeric protons and citric acid methylene protons of interest are in relatively uncluttered regions of the spectrum, and therefore their identification and quantitation is achievable. Indeed, if the selection of the integration ranges is rigorous and consistent with those employed at the high field, the integration of the signals corresponding to anomeric protons of sucrose, glucose, and fructose, as well as the citric acid methylene protons, allows us to apply the methodology developed previously [25] to predict the acceptability of the mandarin samples (Table 1).

The sweetness/citric acid ratio of the samples determined at the two frequencies considered had high correlation (R^2^ > 0.99, Figure 2), showing the equivalence of both systems and their fitness for the intended purpose of the method.

It is then possible to study the correlation between the acceptability of the mandarin samples determined by consumers and the sweetening power/citric acid ratio obtained using the 60 and 400 MHz systems (Figure 3). Using this set of data, a linear regression model with an R^2^ of 0.94 and an RMSE of 0.35 was obtained using data recorded at 400 MHz. The corresponding regression parameters of the linear model derived using sugar and citric acid concentrations determined with the 60 MHz instrument were 0.96 and 0.29, indicating that acceptability prediction models of similar quality were obtained regardless of the instrument employed in their development.

In addition, the correlation between the predicted acceptability using both models was very high (R^2^ > 0.99), further proving the equivalence of the models derived from the two instrumental systems (Figure 4).

It is worth pointing out that although models derived from data at 400 and 60 MHz are of the same predictive quality, special attention is needed when identifying and integrating data in the low-field instrument. As stated earlier and shown in Figure 1, there is considerable signal overlap in the 3.00 to 4.30 ppm region and expertise is required to assign peaks and process these spectra accurately.

Conversely, and due to the lower resolution of low-field instruments, the variations in the chemical shifts of sugar signals with pH have less impact on spectra recorded at 60 MHz [30]. This makes spectral referencing and alignment simpler in these instruments.

## 4. Conclusions

As demonstrated above, low-field NMR systems can be employed in the development of consumer acceptability prediction models that have identical quality to those derived from high-field NMR data. The lower purchase and running costs of benchtop spectrometers makes these chemometric-based tools more accessible for routine inclusion in fruit breeding programs, such as the Uruguayan Programa Nacional de Investigación en Producción Citrícola. Furthermore, the continuing advances in benchtop NMR instruments, which include the implementation pure shift pulse sequences, solvent suppression techniques, and multidimensional and multinuclear methods, will facilitate their application to other fields of food analysis and metabolomics.

## Figures and Tables

**Figure 1 foods-11-02384-f001:**
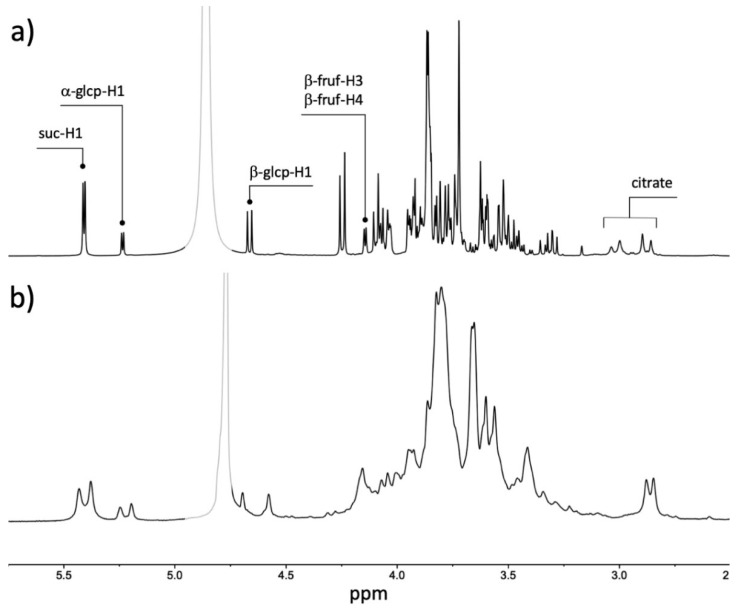
Comparison of ^1^H spectra of the aqueous extract of mandarin variety B475B obtained with 400 and 60 MHz spectrometers ((**a**) and (**b**), respectively). Resonances employed in the estimations are annotated in the 400 MHz spectrum. The grayed-out region in both spectra corresponds to the residual HDO peak.

**Figure 2 foods-11-02384-f002:**
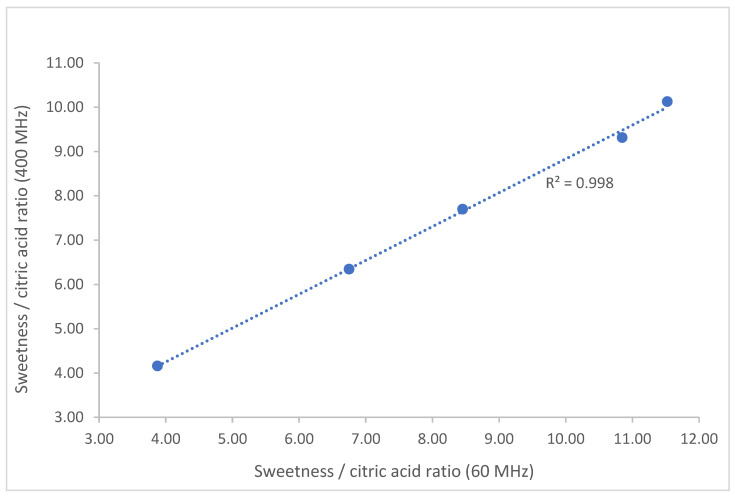
Correlation between the measurement of the sweetness/citric acid ratio of the samples obtained at 400 and 60 MHz.

**Figure 3 foods-11-02384-f003:**
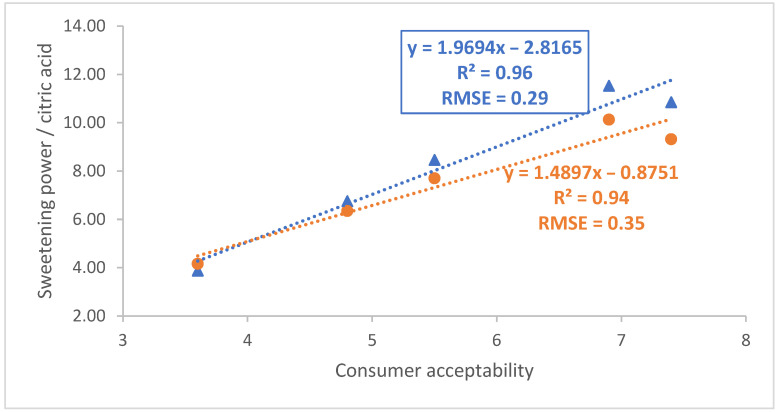
Correlation between consumer acceptability and sweetening power/citric acid determined using 60 and 400 MHz data (blue triangles and orange circles, respectively).

**Figure 4 foods-11-02384-f004:**
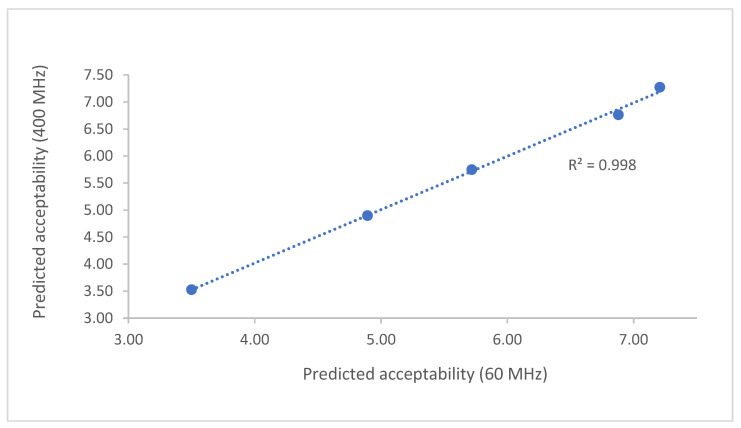
Correlation of the predicted acceptability using the models obtained with both instruments.

**Table 1 foods-11-02384-t001:** Results of the sensory evaluation (acceptability), sweetening power/citric acid ratio, predicted acceptability using the model and RMSE of the prediction of each model (60 and 400 MHz data).

**Variety**	Acceptability	60 MHz			400 MHz		
		Sweetening Power/Citric Acid *	Predicted Acceptability	RMSE	Sweetening Power/Citric Acid *	Predicted Acceptability	RMSE
B475B	7.4	10.85	6.88	0.29	9.31	6.76	0.35
F7P3	6.9	11.53	7.21		10.12	7.27	
B475A	5.5	8.45	5.72		7.70	5.75	
B79	4.8	6.75	4.89		6.34	4.90	
M16	3.6	3.88	3.50		4.16	3.53	

* The reported values correspond to the average of 5 replicates. The RSD was less than 20% for all the varieties analyzed.

## Data Availability

The data presented in this study are available on request from the corresponding author.

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
