# Peer review of "Predicting Mandarin Fruit Acceptability: From High-Field to Benchtop NMR Spectroscopy"

_foods, 2022, doi:10.3390/foods11162384_

Round 1
Reviewer 1 Report
In this paper, high-field and low-field (benchtop) NMR spectroscopy were used to predict mandarin fruit acceptability. The results show that the performance of prediction models established by low field NMR is comparableto that of high field NMR. The experimental results of the paper are good, but the innovation of the paper needs to be improved, and the experimental design is relatively simple. It is suggested to further improve the experimental design and papers.
Author Response
Response to Reviewer 1
In this paper, high-field and low-field (benchtop) NMR spectroscopy were used to predict mandarin fruit acceptability. The results show that the performance of prediction models established by low field NMR is comparable to that of high field NMR. The experimental results of the paper are good, but the innovation of the paper needs to be improved, and the experimental design is relatively simple. It is suggested to further improve the experimental design and papers.
While this reviewer indicates that the “experimental results are good”, he comments that the experimental design is relatively simple and suggests to improve this and -we assume- the paper in general. We believe that one of the key aspects of the work is the simplicity of the experimental design, as it only involves comparing results obtained at high- and low‑field to develop an accurate consumer acceptability model for mandarins. As for improving the paper, we would have appreciated some specifics on how to achieve that.
Reviewer 2 Report
The authors expand on previous work on an NMR-based approach to predict mandarin fruit acceptability. The prediction relies on a calibration involving the measurement of sugar and citric acid signals. Having previously presented results based on a 400MHz spectra, the authors translated this to using NMR spectra acquired on a 60MHz instrument. Here, they present a comparison of the accuracy of these two sets of measurements and of their ability to predict mandarin fruit acceptability.
The work presented is based on a very low number of measurements (5 samples) but suffices to demonstrate the transferability of the approach from high-field to low-field NMR instrumentation, which would be key to it being used in practice.
The manuscript is very well-written and the results clearly presented. There was only one aspect that I felt was not described adequately, namely the integration of each of the signals that are being used to predict fruit acceptability. Yet this is the main challenge (and novel aspect) for transitioning from using the 400MHz to the 60MHz spectra. No detail is given about this. On p.4 the authors state it is straightforward - this may be true for sucrose and alpha-glucose, but not so much for beta-glucose, beta-fructose, and citrate. Indeed on p.5 the authors state that “expertise is required to assign peaks and process these spectra accurately". Here I suggest this step should be accurately described (either graphically on a spectrum, or a written description) to highlight what is novel in the work and allow for it to be replicated. Given this report is based on only 5 samples, an indication of whether difficulties may be encountered during peak integration, and the effect of this, would also be useful.
I also suggest the following minor additions:
- Provide the measurement duration. This would be a useful addition as the readership will not be NMR experts. It would fit in the last paragraph on p5 (concluding on the comparison between the two systems).
- P2 one-before-last paragraph. Provide a reference for alignment of the spectra using the derivative method
- Equation 1. Specify what n corresponds to.
The manuscript is otherwise satisfactory as it stands.
Author Response
The work presented is based on a very low number of measurements (5 samples) but suffices to demonstrate the transferability of the approach from high-field to low-field NMR instrumentation, which would be key to it being used in practice.
Although the model was developed with only five varieties of mandarins, each of the samples was analyzed five times (five genuine replicates). This is now clarified in the Materials and Methods section.
The manuscript is very well-written and the results clearly presented. There was only one aspect that I felt was not described adequately, namely the integration of each of the signals that are being used to predict fruit acceptability. Yet this is the main challenge (and novel aspect) for transitioning from using the 400MHz to the 60MHz spectra. No detail is given about this. On p.4 the authors state it is straightforward - this may be true for sucrose and alpha-glucose, but not so much for beta-glucose, beta-fructose, and citrate. Indeed on p.5 the authors state that “expertise is required to assign peaks and process these spectra accurately". Here I suggest this step should be accurately described (either graphically on a spectrum, or a written description) to highlight what is novel in the work and allow for it to be replicated. Given this report is based on only 5 samples, an indication of whether difficulties may be encountered during peak integration, and the effect of this, would also be useful.
This is an important point, and we agree with the reviewer that the word “straightforward” can be misleading, and we replaced this with “achievable” (line 141, page 4). Indeed, the integration regions have to be selected judiciously and consistently for the method to perform correctly. These integration regions are now clearly stated in the Materials and Methods section.
I also suggest the following minor additions:
- Provide the measurement duration. This would be a useful addition as the readership will not be NMR experts. It would fit in the last paragraph on p5 (concluding on the comparison between the two systems).
The duration of the experiment on each system was added in the Materials and Methods section.
- P2 one-before-last paragraph. Provide a reference for alignment of the spectra using the derivative method
The reference to this method was incorporated (reference 27, Alamprese, 2018).
- Equation 1. Specify what n corresponds to.
“n” corresponds to each of the studied sugars. This is now detailed in the manuscript.
Reviewer 3 Report
The authors comprehensively describe the application of a low field NMR method in comparison to a previously developed high field method to determine the acceptability of mandarin fruits.
I have not much to criticize, the paper is well described and the conclusions are based in the data.
The following corrections could be implemented
Line 76: the botanical name of mandarins could be specified
Lines 82, 84 and throughout: for all suppliers, the cities should be stated
Figure 1: the greyed out region is not well displayed. Could this be colored instead of greyed?
Author Response
Response to Reviewer 3
- Line 76: the botanical name of mandarins could be specified.
The mandarin varieties mentioned in the article are hybrids developed by the National Citrus Program of Uruguay. The data of the parents from which these hybrids descend are detailed in the article cited in the methodology (reference 25, Migues et al., 2021).
- Lines 82, 84 and throughout: for all suppliers, the cities should be stated.
The cities of the suppliers were added to the manuscript.
- Figure 1: the greyed out region is not well displayed. Could this be colored instead of greyed?.
The grayed-out region corresponds to the residual solvent signal and is not relevant. We believe that coloring this signal will not add much to the manuscript and prefer to leave it as it is.
Round 2
Reviewer 1 Report
This paper shows that high-field and low-field (benchtop) NMR spectroscopy can both obtain good results for mandarin fruit acceptability prediction. For the revised paper, the quality and details of the paper have been improved.
For results in Table and figures, the predicted values are the average of 5 replicates. It is better to used the predicted values of 5 replicates? not the average ?
Author Response
As for the quality of the English and wording of the manuscript, in the first round of evaluation this reviewer indicated that only minor spellchecking corrections were needed, but now calls for extensive editing. This comment is confusing, particularly considering that a native English speaker coauthored the manuscript and revised the final version. We therefore believe that, unless specific instructions are provided by the reviewer or the editorial office, the wording and grammar are more than acceptable, and the English language does not need any improvement.
Regarding the way the calculation was performed, the reviewer's suggestion is not clear. We performed the measurements as we did in our previous already published work. We determined the sugars and citric acid in each of the 5 replicates. These values were averaged for each compound and then the sweetening power/citric acid ratio was calculated using the sweetness values of each sugar (which it is also an averaged value). Therefore, a single sweetening power/citric acid value is obtained for each variety, and this is used to correlate with the acceptability value of the mandarin varieties. Mathematically, it is not correct to perform the sweetening power correction for each replicate and then average it, as well as to determine a correlation with acceptability using the individual values of each replica with a single value of acceptability determined by consumers since it also corresponds to an average of 100 individual determinations. That is why the model was generated as described above.